# Could Adverse Effects of Antibiotics Due to Their Use/Misuse Be Linked to Some Mechanisms Related to Nonalcoholic Fatty Liver Disease?

**DOI:** 10.3390/ijms25041993

**Published:** 2024-02-06

**Authors:** Giovanni Tarantino, Vincenzo Citro

**Affiliations:** 1Department of Clinical Medicine and Surgery, Medical School of Naples, Federico II University, 80131 Naples, Italy; 2Department of General Medicine, Umberto I Hospital, Nocera Inferiore (SA), 84014 Nocera Inferiore, Italy; v.citro@libero.it

**Keywords:** antibiotics, use/misuse, food residues, microbiome

## Abstract

Nonalcoholic fatty liver disease, recently re-named metabolic dysfunction-associated steatotic fatty liver disease, is considered the most prevalent liver disease worldwide. Its molecular initiation events are multiple and not always well-defined, comprising insulin resistance, chronic low-grade inflammation, gut dysbiosis, and mitochondrial dysfunction, all of them acting on genetic and epigenetic grounds. Nowadays, there is a growing public health threat, which is antibiotic excessive use and misuse. This widespread use of antibiotics not only in humans, but also in animals has led to the presence of residues in derived foods, such as milk and dairy products. Furthermore, antibiotics have been used for many decades to control certain bacterial diseases in high-value fruit and vegetables. Recently, it has been emphasised that antibiotic-induced changes in microbial composition reduce microbial diversity and alter the functional attributes of the microbiota. These antibiotic residues impact human gut flora, setting in motion a chain of events that leads straight to various metabolic alterations that can ultimately contribute to the onset and progression of NAFLD.

## 1. Introduction

NAFDL, recently re-named metabolic dysfunction-associated steatotic fatty liver disease (MASLD), even though the differences between the old and new acronym are negligible, is considered the most prevalent liver disease all over the world. We use the term NAFLD instead of MASLD throughout this text in light of the fact that the previous literature sources which we refer to use the old acronym [1]. What is more, the diagnostic terms “fatty liver”, “NAFLD”, and “NASH” were not associated with discomfort for the majority of patients. Although 26% of patients reported stigma related to overweight/obesity, only 8% disclosed a history of stigmatisation or discrimination due to NAFLD [2].

The risk factors of NAFLD are multiple and not always clearly defined, ranging from metabolic alterations (dyslipidaemia) to genetic/epigenetic traits [3], lifestyle modifications, including smoking [4], and even viral aetiology [5]. Obesity, mainly abdominal obesity, is one of the most important drivers [6]. Furthermore, patients with the severe form of NAFLD, i.e., nonalcoholic steatohepatitis (NASH) and obesity showed worsening of fibrosis versus those without obesity [7]. Still, sarcopenia is associated with increased risks of NAFLD and advanced fibrosis, independent of obesity or metabolic control [8]. Obviously, type 2 diabetes mellitus (T2DM) plays a determinant role, even though most clinical investigators suggest that NAFLD appearing first leads ultimately to the development of T2DM [9]. 

The mechanisms underlying NAFLD are even more complicated, including gene variants such as patatin-like phospholipase domain-containing protein 3, transmembrane 6 superfamily member 2, hydroxysteroid 17-beta dehydrogenase 13, membrane-bound O-acyltransferase domain-containing 7, and glucokinase regulator [10]. A key role is played by dysfunctional adipocytes, with the involvement of the mesenteric adipose tissue, adipokines such as adiponectin, the food intake hormone (leptin) and leptin resistance, as well as of another adipose tissue’s hormone, i.e., resistin [11]. It has been ascertained that insulin resistance (IR) is central to the onset and progression of NAFLD. Both innate and recruited immune cells, such as macrophages and T-cells, mediate the development of IR. Infiltrated macrophages in obese adipose tissue undergo a phenotypic switch from alternative M2 macrophages to classical M1 macrophages. Consequently, modifying the polarisation of resident and recruited macrophage/Kupffer cells is expected to pave the way to new therapeutic approaches in NAFLD [12]. Lipotoxicity is an important downstream signalling event. Free fatty acids (FFAs) activate apoptosis, including the up-regulation and increased number of death receptors such as Fas and TRAIL receptor 5, at the level of the plasma membrane, lysosomal permeabilisation, and endoplasmic reticulum stress, both coupled to mitochondrial dysfunction and in turn activating the mitochondrial pathway of apoptosis. Furthermore, FFAs set TLR4 signalling in motion, resulting in the up-regulation of several pro-inflammatory cytokines. Finally, other lipids such as free cholesterol and ceramide induce mitochondrial dysfunction and initiate the mitochondrial pathway of apoptosis [13]. Chronic low-grade inflammation represents a further mechanistic level of NAFLD. Inflammatory mediators that are biosynthesised in the liver and increased in NAFLD patients include C-reactive protein, interleukin (IL)-6, fibrinogen, and plasminogen activator inhibitor-1. Recent data lend credence to the fact that hepatic steatosis triggers IκB kinase-beta and nuclear factor-κB. Among the inducible transcription factors that control inflammatory gene expression, nuclear factor-κB plays a central and evolutionarily conserved role in coordinating the expression of various soluble pro-inflammatory mediators (cytokines and chemokines) and leukocyte adhesion molecules, as reviewed in [14]. Recent clues suggest that endoplasmic reticulum stress is involved in the development of lipid droplets and subsequent generation of reactive oxygen species (ROS) in the progression to NASH. The molecular processes caused by the disruption of endoplasmic reticulum homoeostasis, described as unfolded protein response, are associated with membrane biosynthesis, insulin action, inflammation, and apoptosis [15]. Gut microbiota dysbiosis significantly contributes to the pathogenesis and severity of NAFLD. Actually, relatively higher abundances of the genera *Fusobacteria*, and lower abundances of *Oscillospira* and *Ruminococcus* of *Ruminococcaceae* and *Coprococcus* of *Lachnospiraceae*, have been found in NAFLD patients, as well as other bacterial species such as *Proteobacteria*, *Escherichia*, and *Enterobacteria.* Also, *Bacteroides* were shown to be more frequent in patients with NASH. Another study showed the lower diversity of microbiota in the faeces of children with NAFLD and an increased number of *Prevotellacopri.* Finally, a low alpha bacterial diversity was linked with severe liver fibrosis, as reviewed in [16]. Other pathogenic factors and pathological mechanisms known to be involved in the complex disease progression of NAFLD involve programmed cell death include (j) autophagy of fat, referred to as lipophagy, that when impaired can lead to fat accumulation mediated by osteopontin, reduced levels of glycine N-methyltransferase, and phosphorylation of Jumonji-D3; (jj) ferroptosis, characterized by cytological changes that include reduced cell volume and increased mitochondrial membrane density, identified by iron dependence and lipid peroxidation; (jjj) apoptosis/necroptosis, whose key molecules include mixed lineage kinase domain-like and the receptor-interacting protein (RIP) protein kinase family members RIPK1 and RIPK3, with tumour necrosis factor receptor 1-mediated signal transduction being an example of the conversion between apoptosis and necroptosis; and (jjjj) pyroptosis, which may occur through the classic caspase 1-dependent pathway and the nonclassical caspase 4/5 (mouse caspase-11) pathway. All the above pathways are detailed in [17]. Mitochondrial dysfunction is a further molecular process leading to NAFLD due to the increased flux of FFAs in hepatocytes, which leads to increased mitochondrial fatty acid import and oxidation. Consequently, the overproduction of ROS damages mitochondrial membranes and may result in mitochondrial permeability transition pore formation with the consequent release of mtDNA in cytoplasm, which acts as a danger-associated molecular pattern and may activate the NLR family pyrin domain-containing 3 inflammasome with consequent maturation of the cytokine IL-1 beta and perpetuation of inflammation, with this latter mechanism being common to alcoholic liver disease [18]. Bile acids (BAs), beyond FFAs, hormone receptors and drugs, bind to nuclear receptors (NRs) and modulate their transcriptional activity. In hepatocytes and other cell types in the liver, NRs control multiple metabolic and inflammatory processes that influence the development of NAFLD/NASH [19]. It is essential to evidence that adiponectin is inversely linked to IR [20]. Glucocorticoids bear the potential to drive NAFLD, acting on both liver and adipose tissue. In their fasting state, they are able to mobilise lipids, increasing fatty acid delivery, and in their fed state, they can promote lipid accumulation [21]. Recent studies suggest that telomere shortening can lead to cellular dysfunction by impacting metabolism via a p53-dependent repression of mitochondrial biogenesis and function that results in dysfunctional mitochondria [22]. With NAFLD being more prevalent in the elderly [23], telomere length plays an important role in this very common liver disease in the sense that low length accounts for high NAFLD risk, making it plain that telomere shortening causes senescent cells to accumulate [24], Figure 1.

To zero in on long-lasting exposure to some extrinsic factors, such as the use or misuse of antibiotics as well as their food residues, by surfing PubMed, Scopus, Embase, and Research Gate, we aimed to gather data concerning common mechanisms between NAFLD and the side effects of antibiotics. Consequently, by exploring findings drawn from the available preclinical and clinical studies, and concerning the antibiotics mostly used in human and veterinary practice, we focused our attention on the beta-lactam group, tetracyclines, fluoroquinolones, sulphonamides, and aminoglycosides, lending credence to a new possible risk factor for NAFLD.

## 2. Adverse Effects of Antibiotics

### 2.1. Antibiotics Use and Misuse

Nowadays, the excessive use and misuse of antibiotics is a growing public health threat [25]. To slow the rise of this major health hazard, antibiotics should be utilised only when needed and prescribed in a pertinent way (Table 1) and for the right duration. Other issues include self-medication, self-storage, and non-adherence to antibiotic schedules by patients, whose prevalence is high [26].

On the other hand, studies in hospitals show that many currently used antibiotics are not needed, inappropriate agents are chosen, or the dose is incorrect [27]. In an emergency department, among 272 patients enrolled in the study, 68% with bronchitis and 9% with upper respiratory infections received antibiotics. Physicians were more likely to prescribe antibiotics when they believed that patients expected them, although they were able to correctly identify only 27% of the patients to whom antibiotics should have been administered to [28]. Excessive antibiotic use in cold and flu season is costly and, among other side effects, contributes to antibiotic resistance [29]. In this context, we mention the threat of antibiotic resistance that we can possibly face. In fact, resistant microorganisms are also present in the food chain; they can spread between food-producing animals and humans with serious consequences on the patients’ microbiomes. Furthermore, antibiotics are routinely adopted in agriculture and aquaculture. Finally, waste disposal creates major environmental resistance reserves [30]. Because of the random use of antibiotics as growth promoters and production enhancers in livestock, a significant excess of them is released through the unmodified milk of dairy animals, exerting potentially dangerous effects on human health. Specifically, when comparing published studies on antibiotic residues in milk, concentrations of 36.54% of the β-lactam group, 14.01% tetracyclines, 13.46% fluoroquinolones, 12.64% sulphonamides, and 10.44% aminoglycosides were detected [31]. As an example, in Chittagong, Bangladesh, the average concentrations of amoxicillin residue in local milk, commercial milk, local eggs, and commercial eggs were 9.84 µg/mL, 56.16 µg/mL, 10.46 µg/g, and 48.82 µg/g, respectively, in raw samples, and were reduced to 9.81 µg/mL, 55.54 µg/mL, 10.29 µg/g, and 48.38 µg/g, respectively, after boiling [32]. 

### 2.2. Antibiotics and Microbiome

It is noteworthy to stress that antibiotic impact on gut flora bears a great deal of uncertainty concerning its duration. Some studies emphasise the rapid restoration of previous gut flora. In fact, in 20 healthy individuals, an acute decrease in species richness and culturable bacteria due to antibiotics (azithromycin, levofloxacin, cefpodoxime, and azithromycin plus cefpodoxime), administered orally for five days, was followed by most healthy adult microbiomes returning to pre-treatment species richness after two months, with an altered taxonomy, resistome, and metabolic output, as well as an increased antibiotic resistance burden [33]. Accordingly, in a controlled murine system, gut flora during antibiotic treatment recovered after the transient dominance of resistant Bacteroides and taxa-asymmetric diversity reduction, with a fibre-deficient diet exacerbating microbiota collapse and delaying recovery [34]. Vice versa, a recent study assessed the effect of ciprofloxacin (500 mg twice daily for ten days) or clindamycin (150 mg four times daily for ten days) on the faecal microbiota of healthy subjects for a period of one year compared to placebo. The authors found that exposure to those antibiotics had a notable effect on the diversity of the microbiome, and changes in microbial composition were observed until the last month, with the most pronounced microbial shift at the first one [35]. Another study showed that the effect of ciprofloxacin on the gut flora was intense and fast, with a loss of diversity accompanied by shifts in levels of Bacteroidetes, *Lachnospiraceae*, and *Ruminococcaceae* occurring within three or four days of drug initiation. By one week after the end of each course, communities began to return to their initial state, but the gain was often insufficient [36]. In contrast with the knowledge that the commensal microbiota is normalised a few weeks following withdrawal of the antibiotic treatment, increasing documentation suggests that this is not the case and that specific members of the microbiota may be positively or negatively affected for extended periods of time [37]. Recently, it has been again emphasised that antibiotic-induced changes in microbial composition reduce microbial diversity, alter functional attributes of the microbiota, and determine the selection of antibiotic-resistant strains, making hosts more susceptible to infection with pathogens [38]. As a consequence, in high-income countries, overuse of antibiotics, beyond changes in diet, can lead to a microbiota that lacks the “resilience and diversity” required to set up balanced immune responses [39].

### 2.3. Antibiotics and Obesity

The bacterial flora of the body is extraordinary in its richness of distinct species and strains, even though a reduced number of phyla are generally present in indigenous microbial communities. Authors have long proposed that the disappearance of ancestral indigenous organisms forming the microbiota, which are intimately involved in human physiology, is not entirely beneficial and has deleterious consequences, including obesity. This could happen due to the widespread antibiotic use, but not for this reason alone [40]. In a recent “Perspective”, the authors hypothesise that microbiota disruptions in early life can have long-lasting effects on body weight in adulthood [41]. In terms of gut flora, the ileum contains a moderately mixed flora (10^6^ to 10^8^/g of contents). The flora of the large bowel is dense (10^9^ to 10^11^/g of contents) and is composed predominantly of anaerobes [42]. The dominant beneficial bacteria of the gut flora are Bacteroidetes and Firmicutes. The relative proportion of Bacteroidetes is decreased in obese people when compared with lean people, with this proportion increasing with weight loss [43]. Firmicutes and Bacteroidetes potentially mediate IR through modulation of GLP-1 secretion in obesity. Unexpectedly, by altering gut microbiota, vancomycin and bacitracin (potent antimicrobial agents used for the treatment of Clostridium difficile) improve IR via GLP-1 in diet-induced obesity [44]. Accordingly, apart from lifestyle changes, medical practices can also change the prevalence of Bacteroidetes and Firmicutes in human populations. Specifically, preclinical studies evaluating early antibiotic exposure have shown that reductions in the population size of specific microbiota, such as Lactobacillus, Allobaculum, Rikenellaceae, and Candidatus Arthromitus, are related to successive adiposity. In keeping with the previous results, meta-analyses of human studies evidenced an association between antibiotics and the subsequent development of obesity, particularly as a consequence of early exposure in life, as a result of alterations in the diversity of the gut microbiota [45]. Beyond repeated antibiotic treatments in clinical settings, their low-dosage intake from food could be another contributing factor to dysbiosis [46].

### 2.4. Antibiotics and Dyslipidemia

There are many pieces of research dealing with the relation between intestinal flora and cholesterol metabolism in human beings. As a consequence, the alteration or disruption of the microbiome during antibiotic therapy may be able to increase serum cholesterol. Very recently, hypercholesterolemic Apolipoprotein E knockout mice were treated with oral, largely non-absorbable antibiotics. It was found that antibiotics augmented levels of cholesterol, as measured when animals were fed normal chow. Interestingly, the increased concentrations of cholesterol were noted a few days after treatment starting and were reversible after terminating antibiotic administration with the restoration of intestinal bacteria. Finally, gene expression analyses confirmed that the increased intestinal cholesterol uptake was due to antibiotics in the fed state [47]. Researchers have evidenced that amoxicillin treatment acts by decreasing ApoA-I secretion and transcription, the main protein constituent of cholesterol-HDL, suggesting that a decrease in peroxisome proliferator-activated receptor alpha (PPAR-α), i.e., PPARα transactivation, is a downstream molecular event at the basis of the amoxicillin-induced effects on both hepatic and intestinal ApoA-I expression [48]. Lower serum Apo A1 levels were related to NAFLD in normal-weight Korean participants [49]. Consistent with previous observations on this cholesterol-HDL-associated primary protein, data evidenced that the three PPARs (PPARA, PPARD, and PPARG) could boost the expression and molecular transportation of Apo A1. Pathway analysis showed that ApoA1 serves as a hub protein connecting PPARs and NAFLD through a beneficial modulation of 16 out of 21 NAFLD upstream regulators [50].

### 2.5. Antibiotics and Type 2 Diabetes Mellitus

Evidencing a geographical correlation between rates of increase in the prevalence of T2DM and a model using only local rates of outpatient fluoroquinolone prescription (local rates of increase in the prevalence of obesity, and local rates of population growth as predictor variables) in each USA state, recent data are consistent with fluoroquinolone exposure predisposing an individual to develop T2DM, with a probability that relies upon factors that contextually lead to an increase in obesity [51]. The last finding is supported by the advice that fluoroquinolones must be used with great caution among diabetic patients [52]. Recent data could reinforce the likelihood that antibiotics exposure increases T2DM risk. In fact, higher odds ratios (ORs) for T2DM were seen with narrow-spectrum and bactericidal antibiotics (OR 1.55 and 1.48) compared to broad-spectrum and bacteriostatic antibiotics (OR 1.31 and 1.39), respectively. However, these findings could indicate an increased demand for antibiotics due to the augmented risk of infections in such patients [53]. The widespread use of antibiotics is associated with a rise not only in T2DM occurrence, but also in inflammatory bowel disease, coeliac disease, eosinophilic oesophagitis, and type 1 diabetes [54]. Again, it should be stressed that, beyond obesity, dyslipidaemia and T2DM are risk factors for NAFLD [55].

### 2.6. Antibiotics and Spleen

Recent data suggest that splenectomy in male C57BL/6 mice leads to the abnormal composition of the gut microbiota and the short chain fatty acids (SCFAs) (in faecal samples, succinic acid and lactic acid decreased, while propionic acid and n-butyric acid increased) [56]. Thus, the spleen–gut–microbiota axis plays a crucial role in regulating immune homeostasis. Higher SCFAs are characteristic of NASH [57]. Microbiome depletion, induced by the administration of an antibiotic cocktail comprising ampicillin, neomycin sulphate, and metronidazole, led to a significant reduction in spleen weight and consistent alterations in splenic functions, including the percentage of neutrophils, natural killer cells, macrophages, and CD8+ T-cells, over a 14-day period [58].

Recent results in healthy BALB/c mice, an albino, laboratory-bred strain of the house mouse from which a number of common sub-strains are derived, showed that antibiotic treatment with penicillin, kanamycin, and streptomycin for one week determined changes in the composition of the intestinal microbiota, impacted the population of lymphocytes in splenocytes, and altered the immune response. Specifically, Firmicutes were conspicuous in the control group, whereas Bacteroidetes predominated in the antibiotic-treated group, as determined by metagenomic analysis. Accordingly, the Firmicutes/Bacteroidetes ratio is believed to be a main characteristic of obesity. In the antibiotic-treated group, CD3+ cells decreased, CD19+ cells increased, and genes encoding interferon-gamma, IL-6, and IL-13 significantly decreased in splenocytes treated with concanavalin A [59].

## 3. The Connection between Antibiotics and Pathomechanisms of NAFLD

### 3.1. The Link between Antibiotics and NAFLD

In a nationwide case–control study, 2584 Swedish adults with histologically proven NAFLD, of which 56.6% had simple steatosis, 14.8% had steatohepatitis, and 26.8% had non-cirrhotic fibrosis, diagnosed from January 2007 to April 2017, were included and matched to ≤5 12,646 controls for age, sex, calendar year, and county of residence. Antibiotic use was found to be a risk factor for incident NAFLD, especially in individuals without metabolic syndrome. The risk was very significant for fluoroquinolones and remained robust in sibling comparisons with whom individuals shared genetic and early environmental susceptibilities [60].

Concerning the likely mechanisms, very recent data show that intestinal barrier dysfunction in diet-induced NAFLD in male C57BL/6J mice, either pair-fed a liquid control diet or fat- and fructose-rich diet +/− antibiotics (ampicillin/vancomycin/metronidazole/gentamycin) for seven weeks were not based on changes in intestinal microbiota but rather on altered intestinal nitric oxide (NO) homeostasis induced by fructose [61]. It has been shown that the development of NAFLD with starting inflammation was linked to impaired intestinal barrier function, increased NO levels, and a loss of arginase activity in the small intestines of female C57BL/6J mice that were pair-fed with a liquid control diet or a fat-, fructose-, and cholesterol-rich diet for eight weeks [62]. Treating mice with a variety of antibiotics (cefoperazone, clindamycin, and vancomycin) to create distinct microbial environments in the large intestine led to a significant loss of secondary BAs, such as deoxycholate, lithocholate, ursodeoxycholate, hyodeoxycholate, and ω-muricholate. These changes were related to the loss of specific microbiota community members, such as the Lachnospiraceae and Ruminococcaceae families [63]. Compared to controls, mice with the progressive form of NAFLD, i.e., NASH, had lower concentrations of secondary BAs in their portal blood and bile, while systemic BA concentrations were not significantly altered [64].

### 3.2. Other Molecular Initiation Events

It is not surprising that antibiotics act not only on the bacteria of exogenous infections, but also on the mitochondria from the human microbiome, explaining why their use bears various adverse side effects, such as mitochondrial dysfunction [65], increased ROS generation, and decreased ATP production with increased folding of the inner membrane, all of these mechanisms having a role in obesity and T2DM [66]. More and more data indicate that hepatic mitochondrial dysfunction is crucial to the pathogenesis of NAFLD [67]. We have previously underlined that low telomere length accounts for high NAFLD risk. Surprisingly, interesting findings showed that ofloxacin and levofloxacin, quinolones that are largely prescribed, were characterised at low concentrations by decreased telomerase activity, evaluated by absorbance values that were lower than the control cells, indicating a reduction in the rate of cell proliferation [68]. However, it should be stressed that azithromycin in turn attacks senescent cells, efficiently removing almost the totality of them, thus acting as a potent senolytic agent [69]. Gut microbiota influence host physiology by epigenetic regulation, ending up in chemical donors for DNA or histone modifications, or modifying enzyme expression and/or activity, or generating host-cell-intrinsic processes that direct epigenetic pathways. SCFAs represent another important group of epigenetically relevant molecules that are exclusively produced by commensal microbes through the fermentation of complex non-digestible carbohydrates and fibre [70]. It is well known that higher SCFA excretion was associated with evidence of gut dysbiosis, gut permeability, and excess adiposity [71]. It is noteworthy that serum concentrations of isobutyrate and methylbutyrate and their involvement in glucose metabolism, and in regulating insulin sensitivity and lipogenesis through diverse pathways, were significantly and negatively correlated with NAFLD severity [72]. 

Circulating meal-associated leptin and ghrelin levels and BMI changed significantly after H. pylori eradication, providing direct evidence that H. pylori colonisation, but also antibiotics, have effects on body morphometry [73]. Novel results showed that gut dysbiosis due to long-term use of systemic antibiotics can impact oral microbiota and aggravate periodontitis. Furthermore, the expression of cytokines related to Th17 was increased, while transcription factors and cytokines related to Treg were decreased in the periodontal tissue [74]. There is a close association between NAFLD and periodontal disease, as confirmed by epidemiological studies, basic research, and immunology, with Th17 being a key molecule for explaining the relationship between these two diseases [75].

## 4. Discussion

The widespread use of antibiotics has become a matter of concern, resulting in their abuse or misuse not only in humans, but also in animals; this veterinary approach has led to the presence of residues in derived foods, such as milk and dairy products. According to 1371 articles in published literature, residues of β-lactam group have been detected in 36.54%, followed by tetracyclines in 14.01%, fluoroquinolones, 13.46%, sulfonamides in 12.64% and aminoglycosides in 10.44% [31].

Fluoroquinolones, third-generation cephalosporins, macrolides, and polymyxins, as well as tetracyclines, aminoglycosides, sulphonamides, and penicillins, are registered for use in poultry in all countries [76]. Heat treatments can reduce the risk of some sulphonamides, tetracyclines, and fluoroquinolones but do not guarantee the complete elimination or degradation of these antibiotic residues present in broiler meat [77]. Antibiotics have been used since the 1950s to control certain bacterial diseases of high-value fruit and vegetables [78].

All these antibiotic residues impact the human gut flora, modifying its diversity and composition and setting in motion a chain of events that leads straight to various metabolic alterations that can ultimately contribute to the onset and progression of NAFLD. A metagenome-wide association study on stools from 218 individuals with atherosclerotic cardiovascular disease showed that their gut microbiomes showed increased abundances of *Enterobacteriaceae* and *Streptococcus* spp., differently from 187 healthy controls, concluding that the gut microbiota could play a role in cardiovascular diseases [79]. What is more, consumption of functional foods rich in phytochemicals as well as probiotics are feasible interventions to ameliorate gut health and reduce plaque burden in patients suffering from cardiovascular disease [80]. The previous finding could have an important clinical aspect. In fact, there is proof that patients with NAFLD are at increased risk for the development of coronary heart disease, which clinically results in increased cardiovascular morbidity and mortality [81]. Unfortunately, not all studies are in support of the increased risk of cardiovascular disease caused by NAFLD. In fact, metabolic dysfunction per se rather than hepatic steatosis explained cardiovascular risk, assessed using SCORE2/ASCVD, in 1903 patients out of 4286 asymptomatic subjects from the SAKKOPI study [82]. This is in line with a previous study in which the authors concluded that the risk of cardiovascular disease should be assessed in the standard way in these patients, and that NAFLD should not be considered as a risk enhancer [83], and another study showing that the diagnosis of NAFLD in the current routine care of 17.7 million patients appears not to be associated with acute myocardial infarction or stroke risk after adjustment for established cardiovascular risk factors [84]. The likely explication is that lifestyle factors such as smoking, sedentary lifestyle with poor nutrition habits, and physical inactivity contribute with identical mechanisms to the development of both cardiovascular disease and NAFLD [85]. Another interesting aspect is that macrolides and rifampin led to a decrease in psoriasis area and severity index score in plaque-type psoriasis, while penicillin revealed no statistically significant improvement in guttate psoriasis [86]. This emphasises the co-presence of psoriasis in NAFLD patients, due to the common role of chronic low-grade inflammation and spleen in these two diseases [87]. Finally, there is a remarkable link between antibiotic side effects and disability. In fact, increasing evidence has shown that some classes of antibiotics can induce muscle weakness, pain, and a feeling of fatigue upon resuming physical activity [88]. Gut dysbiosis and intestinal barrier dysfunction, both affected by antibiotics, appear to be a significant limitation in primary and secondary sarcopenia management, due to enhanced microbial translocation that may negatively affect muscle strength, physical function, and frailty [89]. Again, multiple cross-sectional observational studies have demonstrated that sarcopenia is associated with an increased risk for NAFLD and NAFLD-related advanced fibrosis in NAFLD across ethnicities [90]. The central factors affecting muscle loss include increased levels of ROS, DNA damage, and cell apoptosis [91]. In keeping with the last mechanism, cells treated with amoxicillin and clarithromycin showed the formation of apoptotic bodies, and degeneration and detachment of the cells in a dose-dependent manner [92]. As a final observation, there was a positive association between long-term air pollution exposure and NAFLD [93]. Interestingly, air pollution may contribute to antibiotic resistance [94] and IR [95].

## 5. Conclusions

The future of antibiotics necessitates substantial changes in policy, a quantitative understanding of the collective value of these drugs, and investments in alternatives to traditional antibiotics, such as narrow-spectrum drugs, bacteriophages, monoclonal antibodies, and vaccines. All of them should be associated with more effective diagnostic tools. This approach could be useful for the avoidance or reduction of antibiotic side effects, antibiotic resistance, and antibiotic-related metabolic disorders, such as NAFLD. 

## Figures and Tables

**Figure 1 ijms-25-01993-f001:**
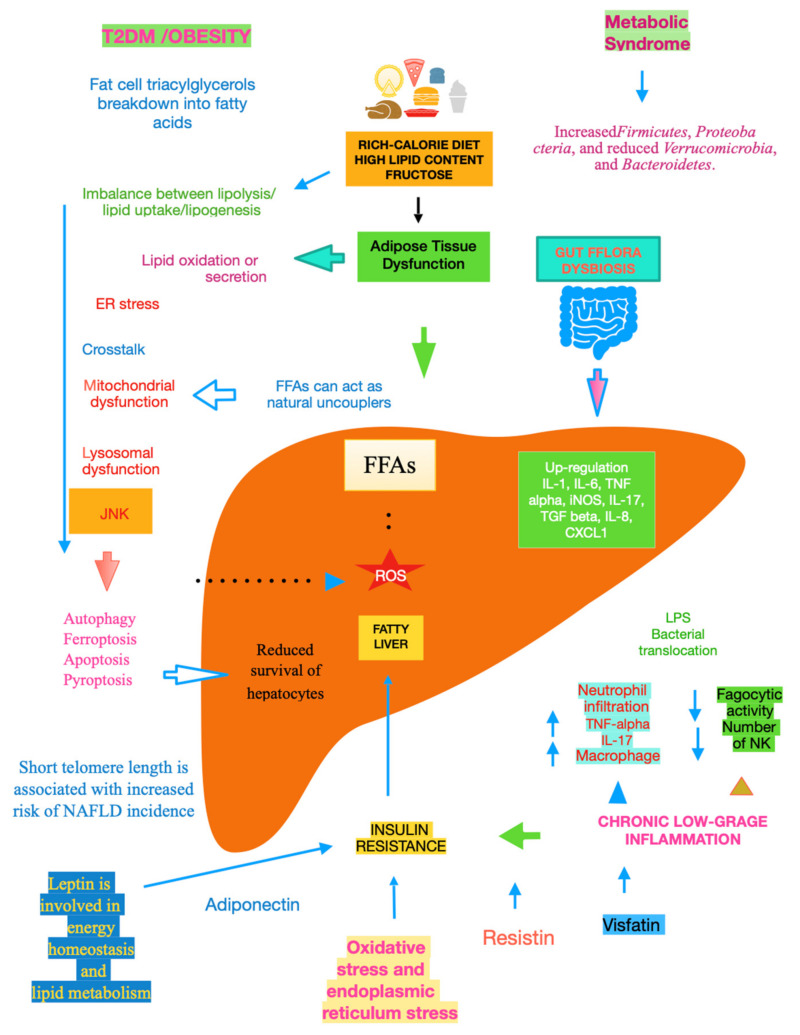
Main mechanisms of NAFLD.

**Table 1 ijms-25-01993-t001:** Common antibiotics used in medical practice with their precise indications.

**AMPICILLIN**
Bacterial meningitis. Sepsis. Endocarditis caused by enterococcal strains.
Gastrointestinal infections by Salmonella. Genito-urinary tract infections.
Prophylaxis for rheumatic heart disease or dental procedures, vaginal hysterectomies, or C-sections. Used in pregnant carriers of group B streptococci to prevent early-onset neonatal infections.
Respiratory infections. Sinusitis. Otitis media. Gonorrhoea if not resistant to penicillin.
Healthcare-associated infections using urinary catheters. Whooping cough to prevent and treat secondary infections.
**TETRACYCLINE**
Moderately severe acne and rosacea.
Prophylactic treatment for infection by bacillus anthracis. Efficacy against Yersinia pestis of bubonic plague.
Treatment and prophylaxis of malaria. Elephantitis filariasis.
Treatment of choice for infections caused by chlamydia (trachoma, psittacosis, salpingitis, urethritis, and L. venereum infection), Rickettsia (typhus, Rocky Mountain spotted fever), brucellosis, and spirochaetal infections (Lyme disease, borreliosis, and syphilis).
**FLUOROQUINOLONES**
Genitourinary infections. Treatment of hospital-acquired infections associated with urinary catheters. In community-acquired infections when risk factors for multi-drug resistance are present or after other antibiotic regimens have failed.
Acute cases of pyelonephritis or bacterial prostatitis where the patient needs to be hospitalised.
Drugs approved for use in children only under narrowly defined circumstances.
The “drugs of choice” in patients with sickle-cell disease with osteomyelitis from salmonella due to their ability to enter bone tissue.
**SULFONAMIDES**
Brucellosis. Nocardiosis. Ulcus molle. Granuloma inguinale.
Typhus. Paratyphoid A and B. Shigellosis. Traveler’s diarrhoea. Kidney and urinary tract infections. Respiratory infections, such as pneumonia and sinusitis, as a second-line drug.
**AMINOGLYCOSIDES**
Infections involving aerobic Gram-negative bacteria, such as pseudomonas, acinetobacter, and enterobacter. Some mycobacteria, including the bacteria that cause tuberculosis, are susceptible.
Their most frequent use is as “empiric” therapy for serious infections such as sepsis, complicated intra-abdominal infections, complicated urinary tract infections, and nosocomial respiratory tract infections.
**MACROLIDES**
Infections due to legionella pneumophila, mycoplasma, mycobacteria, some rickettsia, and chlamydia.
Diseases caused by beta-haemolytic streptococci, pneumococci, and staphylococci.
**METRONIDAZOLE**
Abdominal, soft tissue, gum, and tooth infections. Vaginal infection by trichomonas vaginitis. Bacterial infection by bacterial vaginosis.
Abscesses in the lungs or brain.
Protozoal infections, such as amebiasis. Intestinal infections such as giardiasis. Helicobacter p. eradication.
**CEPHALOSPORINS**
The different generations are effective against different types of bacteria ranging from skin and soft tissue infections to some respiratory ones, and to serious infections such as meningitis or hospital-acquired pneumonia.

## Data Availability

Not applicable.

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
