# Peer review of "Could Adverse Effects of Antibiotics Due to Their Use/Misuse Be Linked to Some Mechanisms Related to Nonalcoholic Fatty Liver Disease?"

_ijms, 2024, doi:10.3390/ijms25041993_

Round 1

Reviewer 1 Report (Previous Reviewer 1)

Comments and Suggestions for Authors

The manuscript titled „Could antibiotics use/misuse be an unsuspected cause of NAFLD pandemic and what is beyond microbiome?’’ is a review which require major improvements.

1.       The title does not correspond to the content of the manuscript. That manuscript is rather focused on adverse effects of antibiotics. It should be completely modified, more focused on the mechanisms related to NAFLD.

2.       The pathomechanism of NAFLD should be described in detail in Introduction.

3.       Why do the authors assume that antibiotics contribute to the increased incidence of NAFLD? – Provide significant data in the last paragraph of the Introduction, please.

4.       Chapter 1 – is there any data indicating which group of antibiotics is more involved in the formation of NAFLD? – please provide it.

5.       Chapter 2 presents adverse effects of antibiotics – it is not strongly associated with formation of NAFLD. Correct it, please

6.       Chapter 3.2 may be included into Chapter 2

7.       Chapter 3 should contain the connection between antibiotics and pathomechanism of NAFLD and conclusion

8.       Chapter 4, in my opinion, is not associtated with the subject of that paper.

9.       There are many grammatical mistakes – Professional Language Service is required

10.   Many abbreviations are not explain: FFAs, Ors, T2DM, BAs, NO

11.   BALB/c mice – a kind of modification should be provide

I do not recommend that paper in the present form. In my opinion, it requires general correction and after that it should be re-checked.

Comments on the Quality of English Language

Therea are many grammatical mistakes. English should be checked by Professional Language Service

Author Response

Dear Editor, 

A according to my co-author, I have accepted all the comments  made by referees, whom I warmly thank for their  precious suggestions.

G Tarantino, MD 

REVIEWER 1

The manuscript titled „Could antibiotics use/misuse be an unsuspected cause of NAFLD pandemic and what is beyond microbiome?’’ is a review which require major improvements.

COMMENT (C)

ANSWER (A)

C1.       The title does not correspond to the content of the manuscript. That manuscript is rather focused on adverse effects of antibiotics. It should be completely modified, more focused on the mechanisms related to NAFLD.

A: The title was modified according to the suggestion, in….Could  adverse effects of antibiotics due to their use/misuse be linked to some mechanisms related to NAFLD ?

C2.       The pathomechanism of NAFLD should be described in detail in Introduction.

A: The mechanisms were described in details on the basis of the  previously reported references.

C3.       Why do the authors assume that antibiotics contribute to the increased incidence of NAFLD? – Provide significant data in the last paragraph of the Introduction, please.

A: In accordance with the comment, it is imprecise the concept that… use and mainly misuse of antibiotics, mainly as food residues, are the cause of the increased incidence of NAFLD (pandemic), thus  the final part of the Introduction section has been modified.

The new text is….. Now, zero in on  a long lasting exposure of some extrinsic factors, such as use or misuse of antibiotics as well as their food residues, surfing PubMed, Scopus, Embase, Research Gate, we aimed at gathering  data concerning common mechanisms between  NAFLD and side-effects of antibiotics.

C4.       Chapter 1 – is there any data indicating which group of antibiotics is more involved in the formation of NAFLD? – please provide it.

A: This sentence was added at the end of the Chapter 1 

 Consequently, exploring findings drawn from  available preclinical and clinical studies, concerning the mostly used in human practice and in veterinary antibiotics, we focused our attention on beta-lactam group, tetracyclines, fluoroquinolones, sulfonamides and aminoglycosides lending credence to  a new  possible risk factor for NAFLD.

C5.       Chapter 2 presents adverse effects of antibiotics – it is not strongly associated with formation of NAFLD. Correct it, please

A:  The hiding of Chapter 2 was changed in… Adverse effects of antibiotics, while that of Chapter 2.1. was changed in...Antibiotics use and misuse

C6.       Chapter 3.2 may be included into Chapter 2

A: It was done

C7.       Chapter 3 should contain the connection between antibiotics and pathomechanism of NAFLD and conclusion

A:  The Chapter 3 was modified as suggested in… The connection between antibiotics and pathomechanisms of NAFLD

C8.       Chapter 4, in my opinion, is not associated with the subject of that paper.

A. The Chapter 4 was deleted with related references

C9.       There are many grammatical mistakes – Professional Language Service is required

A: The English language was ameliorated amending  the grammatical mistakes and typographical errors

C10.   Many abbreviations are not explain: FFAs, Ors, T2DM, BAs, NO

A. The acronyms were clarified, I.e FFAs, Ors,T2DM, BAs and NO

C11.   BALB/c mice – a kind of modification should be provide

A:  BALB/c mice is an albino, laboratory-bred strain of the house mouse from which a number of common substrains are derived. This sentence was added.

C12. I do not recommend that paper in the present form. In my opinion, it requires general correction and after that it should be re-checked. Comments on the Quality of English Language:There are many grammatical mistakes. English should be checked by Professional Language Service

A:  Authors improved the readability of manuscript, amending grammatical mistakes and typographical errors.

Reviewer 2 Report (Previous Reviewer 2)

Comments and Suggestions for Authors

The theme of the narrative review prepared by the authors is beyond doubt.

However, the re-submitted manuscript has been re-framed without taking into account the recommended template provided in the Authors' Guide.

The title of the manuscript needs revision and clarification.

In the Introduction section, it is necessary to add the existing problems that prompted the authors to prepare this manuscript. What was their goal and search strategy?

The title of section 2 is missing, only subsections 2.1, 2.2 and so on are presented.

The structure of the Table (line 103) is difficult to understand.

The name of subsection 2.2 needs correction, the role of this mechanism in the development of the disease in question has not been disclosed by the authors.

The Discussion section needs a major revision.

Comments on the Quality of English Language

Moderate technical and stylistic correction is necessary

Author Response

REVIEWER 2

COMMENT (C)

ANSWER (A)

C: The theme of the narrative review prepared by the authors is beyond doubt.

A: Authors are grateful to the reviewer for his/her appreciation

C: However, the re-submitted manuscript has been re-framed without taking into account the recommended template provided in the Authors' Guide.

A: Authors frankly apologise for this inconvenience, hoping in the help of the Assistant Editor.

C: The title of the manuscript needs revision and clarification.

A: The title was modified, also in accordance with the other reviewer, in….Could  adverse effects of antibiotics due to their use/misuse be linked to some mechanisms related to NAFLD ?

 C: In the Introduction section, it is necessary to add the existing problems that prompted the authors to prepare this manuscript. What was their goal and search strategy?

A: These sentences were added to the final part of the Introduction section…

Now, zero in on  a long lasting exposure of some extrinsic factors, such as use or misuse of antibiotics as well as their food residues, surfing PubMed, Scopus, Embase, Research Gate, we aimed at gathering  data concerning common mechanisms between  NAFLD and side-effects of antibiotics. Consequently,  exploring findings drawn from  available preclinical and clinical studies, concerning the mostly used in human practice and in veterinary antibiotics, we focused our attention on beta-lactam group, tetracyclines, fluoroquinolones, sulfonamides and aminoglycosides lending credence to  a new  possible risk factor for NAFLD.

C: The title of section 2 is missing, only subsections 2.1, 2.2 and so on are presented.

A: A new title was added, also according with the other reviewer, i.e.Adverse effects of antibiotics, while that of section 2.1. was changed in ..Antibiotics use and misuse

C: The structure of the Table (line 103) is difficult to understand.

A:  Being not available the numeration on my iMac, unfortunately the line 103 was not found. Anyway, the structure of Table was ameliorated and some mistakes were amended.

C: The name of subsection 2.2 needs correction, the role of this mechanism in the development of the disease in question has not been disclosed by the authors.

A: Subsections’ titles were changed also according to the other reviewer

C: The Discussion section needs a major revision.

A: The Discussion section was revised and amplified in the section of Conclusive remarks and future directions, focusing on sarcopenia, condition present in various diseases that share some common mechanisms with NAFLD and gut dysbiosis  induced by antibiotics, apart apoptosis and air pollution.

Round 2

Reviewer 1 Report (Previous Reviewer 1)

Comments and Suggestions for Authors

The present version of the manuscript is better but some corrections are necessary:

line 56 - remove "et"

line 165 - remove "he"

line 168 - provide bracket

line 205 - use the abbreviation "GLP-1" instead of full name

line 356 - it is better to use "cardiovascular" instead of the abbreviation "CV", see also line 364

Figure, especially the left side, seems to be not finished.

After these correction the manuscript can be accepted.

Comments on the Quality of English Language

The english language is significantly better, although I see some editorial mistakes.

Author Response

Reviewer 1 

The present version of the manuscript is better but some corrections are necessary:

line 56 - remove “et" 

Answer: It was fixed

line 165 - remove “he"

Answer: It was removed

168 - provide bracket

Answer: It was fixed

line 205 - use the abbreviation "GLP-1" instead of full name

Answer: It was fixed

line 356 - it is better to use "cardiovascular" instead of the abbreviation "CV", see also line 364

Answer: It was fixed

Figure, especially the left side, seems to be not finished.

Answer: It was reshaped with the adding of other mechanisms

After  these correction the manuscript can be accepted.

Comments on the Quality of English Language

The english language is significantly better, although I see some editorial mistakes.

I am deeply grateful to you.

G Tarantino, MD

Reviewer 2 Report (Previous Reviewer 2)

Comments and Suggestions for Authors

I thank the authors for their response. The manuscript has been substantially revised by the authors.

I recommend replacing the title of section 4 with "Discussion" and adding a section "Conclusion" (2-3 sentences), where the main findings of this narrative review are relevant. I would like to draw the authors' attention to the fact that the conclusion section does not contain links to previously published articles.

Author Response

Reviewer 2

I thank the authors for their response. The manuscript has been substantially revised by the authors.

I recommend replacing the title of section 4 with "Discussion" and adding a section "Conclusion" (2-3 sentences), where the main findings of this narrative review are relevant. I would like to draw the authors' attention to the fact that the conclusion section does not contain links to previously published articles.

Answer :The section 4, i.e., Conclusive remarks and future direction was split into two subsections, i.e., Future Directions with related references and Conclusion without references.

I am deeply grateful to you,

G Tarantino, MD

This manuscript is a resubmission of an earlier submission. The following is a list of the peer review reports and author responses from that submission.

Round 1

Reviewer 1 Report

Comments and Suggestions for Authors

The manuscript titled: "Could both alcoholic and nonalcoholic liver disease..." is not prepared with editorial guielines. Besides, that manuscript is not professionaly prepared - there is no purpose of the manuscript, methods, and conclusions are not clear.

In my opinion it does not meet the requirements of such prestigous journal.

I do not recommend it for publication.

Comments on the Quality of English Language

English language require professional correction.

Author Response

Reply to the Editor and Reviewers, 

First of all, we would warmly thank the Editor for giving us the possibility to reply and both of Reviewers for the time and the commitment in revising the manuscript.

Secondly, we have accepted all the  comments/suggestions, being sure that each of them was made with constructive spirit, also the harsh ones. Indeed, the  criticism  has prompted us to improve as better as possible the content of our article addressing a major health issue.

Kind regards, 

G Tarantino, MD and Coll.

Comments and Suggestions for Authors from Reviewer 1 

Comment (C). 

The manuscript titled: "Could both alcoholic and nonalcoholic liver disease..." is not prepared with editorial guidelines. 

Answer (A).

The manuscript was deeply revised in its sections, adding the search strategy and modifying the Discussion section that contain the Conclusions and Future Directions. 

C. Besides, that manuscript is not professionally prepared - 

A. We have reduced the content of Introduction, putting a  curb on some pleonastic aspects. Also the Discussion section was re-arranged.

C. there is no purpose of the manuscript, 

A. We have given details on this point, I.e.,  

Aim of the review 

We reviewed evidence on the interference of NAFLD and/or alcoholic liver disease with opioids use/abuse. We show that not only may NAFLD and alcoholic liver disease occur in the context of these drugs  hepatotoxicity, but also that preexisting NAFLD and alcoholic liver disease influences the susceptibility to opioid  hepatotoxicity. 

C. methods, 

A. We have given details on this aspect, I.e., 

Methods

To prepare this narrative review, we interrogated PubMed (https://pubmed.ncbi.nlm.nih.gov/ (accessed on April 2023)), Scopus (https://www.scopus.com/search/form.uri?display=basic&zone=header&origin=#basic (accessed on April 2023)) and Embase (https://www.embase.com/ (accessed on April 2023)) to track recent evidence using the following keywords: opiods, opioids analogs, fentanyl, NAFLD, hepatic steatosis, gut-liver axis, gut microbiome, obesity, metabolic syndrome, alcoholic liver disease, hepatotoxicity, and CYP 450. This search ended up in summing up a number of preclinical and clinical studies present in literature and focusing on inner mechanisms that induce liver damage. To give readers a more comprehensive view of the issue some older pieces of research concerning opioid liver toxicity were also reported to emphasise the initial and deserving interest of scientists towards this issue that unfortunately was lost in time. For this reason, it could be useful to try to draw the attention of physicians and health care authorities on a possible, sometime severe, risk of liver damage subsequent the chronic use of these substances, with fentanyl as a cornerstone, in patients with hidden or full-fledged alcoholic and nonalcoholic liver diseases.

C. and conclusions are not clear.

A.We tried to modify the content of the  Conclusions section

C. In my opinion it does not meet the requirements of such prestigous journal.

A. We wish that, on the basis of the efforts made in the improvement of the manuscript, this opinion could be changed.

I do not recommend it for publication.

Comments on the Quality of English Language

C. English language require professional correction.

A. We made efforts to improve the language all over the text, also amending typos, and sincerely regret for the unintended deviation from some rules of English.

Reviewer 2 Report

Comments and Suggestions for Authors

The topic of the descriptive review is relevant, there is no doubt about it. The manuscript is easy to read, there were no difficulties with understanding its content. However, this manuscript reminds me of a chapter from a monograph. It is known that the basis of the methodology of a descriptive review is the definition of a search strategy (sources and keywords), but the author's search strategy and its algorithm are not entirely clear. The goal that the authors set for themselves when preparing the manuscript is not formed. The search depth is also important, which should not exceed the last 5 years (maximum 10 years). The authors cite earlier publications and outdated theories.

The contents of all tables need to be structured and unnecessary details removed.

The sections "Detection methods" and "Animal models" are very small and need major revision.

It would be useful to add illustrations to improve the perception of this manuscript.

There is no "Discussion" section, in which it is necessary to note the main problems that the authors have revealed. Also, in this section it is important to describe the main differences of this narrative review from previously published ones, as well as its limitations.

The section "Conclusion" needs to be shortened and a reasoned summary of the main findings of the authors. References to previously published works are not allowed in this section. I recommend the authors to move the debatable questions from the "Conclusion" to the new section "Discussion". The content of the "Future directions" section should be moved to the "Discussion" section as well.

The authors did not use the template that is presented in the Authors' Guide on the MDPI website. This makes it difficult to review. The manuscript needs major technical correction.

Comments on the Quality of English Language

The manuscript needs a slight correction of the style of presentation in English.

Author Response

Reply to the Editor and Reviewers, 

First of all, we would warmly thank the Editor for giving us the possibility to reply and both of Reviewers for the time and the commitment in revising the manuscript.

Secondly, we have accepted all the  comments/suggestions, being sure that each of them was made with constructive spirit. Indeed, the  criticism  has prompted us to improve as better as possible the content of our article addressing a major health issue.

Kind regards, 

G Tarantino, MD and Coll.

Comments and Suggestions for Authors from Reviewer 2.

Comment (C). The topic of the descriptive review is relevant, there is no doubt about it.

Answer (A). It is a novel and major health problem according to the respected Reviewer.

 C. The manuscript is easy to read, there were no difficulties with understanding its content. However, this manuscript reminds me of a chapter from a monograph.

A. We tried to perform an easy and readable piece of  research in contrast with the complexity of the issue.

C. It is known that the basis of the methodology of a descriptive review is the definition of a search strategy (sources and keywords), but the author's search strategy and its algorithm are not entirely clear. 

A. It was clarified in methods

C. The goal that the authors set for themselves when preparing the manuscript is not formed. The search depth is also important, which should not exceed the last 5 years (maximum 10 years). The authors cite earlier publications and outdated theories.

A. It was clarified in the Aim section.

C. The contents of all tables need to be structured and unnecessary details removed.

A: Tables were ameliorated, curbing pleonastic pieces of information.

C. The sections "Detection methods" and "Animal models" are very small and need major revision.

A. This part was somehow amplified as pertinently suggested and was put in the Discussion section, even though there is scarcity of specific animal models.

C. It would be useful to add illustrations to improve the perception of this manuscript.

A. It was generated a coloured Figure to show the main mechanisms of opioids liver toxicity.

C. There is no "Discussion" section, in which it is necessary to note the main problems that the authors have revealed. 

A.The Discussion section was obtained by the previous parts of the manuscript that were reorganised.

C. Also, in this section it is important to describe the main differences of this narrative review from previously published ones, as well as its limitations.

C. The section "Conclusion" needs to be shortened and a reasoned summary of the main findings of the authors. References to previously published works are not allowed in this section. I recommend the authors to move the debatable questions from the "Conclusion" to the new section "Discussion". The content of the "Future directions" section should be moved to the "Discussion" section as well.

  1. The suggestions were followed, having the authors found them very pertinent. The reference concerning a previously published article was opportunely put in another section. Some further references were added in relation to sentences added in the text.

C. The authors did not use the template that is presented in the Authors' Guide on the MDPI website. This makes it difficult to review. The manuscript needs major technical correction.

A. We  hope that the Editorial Office will help us.

.

C.  Comments on the Quality of English LanguageThe manuscript needs a slight correction of the style of presentation in English.

A.  A  revision of the English was performed.

Round 2

Reviewer 2 Report

Comments and Suggestions for Authors

The authors have improved the manuscript, but its quality is still low.

It is difficult for me to give specific comments on where and what needs to be corrected, since the authors refused to use the manuscript template.

The contents of the tables are unnecessarily detailed. No changes have been made by the authors.

For example, in the second column of Tables 1, 2, and 3 it is necessary to clearly formulate the type of liver damage and the mechanism, and not describe the findings of the cited authors. In addition, there are no reference numbers in these tables. The authors use abbreviations in the tables, but no Note has been added under the tables where it is necessary to clarify these abbreviations.

The title of the manuscript states "What is the role of microbiome?". However, this aspect is not sufficiently disclosed by the authors, and this problem is not discussed in the Discussion section. These authors are not ready to significantly modify the manuscript, then from the title of the article "What is the role of microbiome?" must be deleted.

Comments on the Quality of English Language

The style of the English language still needs correction.

Author Response

Comments and Suggestions for Authors

Comment (C): The authors have improved the manuscript, but its quality is still low.

Answer (A): We tried to adhere to the following author’s guidelines of IJMS, ie, Reviews offer a comprehensive analysis of the existing literature within a field of study, identifying current gaps or problems. They should be critical and constructive and provide recommendations for future research. No new, unpublished data should be presented. The structure can include an Abstract, Keywords, Introduction, Relevant Sections, Discussion, Conclusions, and Future Directions, with a suggested minimum word count of 4000 words.

All Figures, Schemes and Tables should be inserted into the main text close to their first citation and must be numbered following their number of appearance (Figure 1, Scheme I, Figure 2, Scheme II, Table 1, etc.). All Figures, Schemes and Tables should have a short explanatory title and caption. All table columns should have an explanatory heading. To facilitate the copy-editing of larger tables, smaller fonts may be used, but no less than 8 pt. in size. Authors should use the Table option of Microsoft Word to create tables.

C: It is difficult for me to give specific comments on where and what needs to be corrected, since the authors refused to use the manuscript template.

A: The Discussion section was amplified according to suggestions, adding new sentences.  Anyway, we are very sorry for our misinterpretation of the guidelines, hoping to have ameliorated every section.

C: The contents of the tables are unnecessarily detailed. No changes have been made by the authors.

A: Thank you for this suggestion. Our efforts in curbing the details ended up in eliminating precious data, the unique available in that period, so unfortunately we were forced to maintain many parts of the text.

C: For example, in the second column of Tables 1, 2, and 3 it is necessary to clearly formulate the type of liver damage and the mechanism, and not describe the findings of the cited authors.

A: Data on every article are incomplete on mechanisms and type of damage, thus, the heading is a little bit wrong, and for this, we  agree with the referee that Table 1 should be modified in the second column. To love of truth and respecting the willingness of readers to know as much as possible data on the research, we changed  the sentence “Type of damage and Mechanisms” with “Results from the study”. The content  of the Descriptions was reduced when possible and symbols were added.The references’ number and the year of publication are present in the third column. Also the caption was modified accordingly.

Concerning the Table 2 and 3, because often authors of the article taken into account do nor give  complete pieces of information about some alterations, mainly about related mechanisms, the heading of the second column was changed in “Findings from the study”. Furthermore, the sentences were further polished, adding symbols when necessary.  The references numbers plus year of publication of the articles were displayed in the third column. One caption for each Table was added.

C:  In addition, there are no reference numbers in these tables.

A: Please, see above.

C: The authors use abbreviations in the tables, but no Note has been added under the tables where it is necessary to clarify these abbreviations.

A: An extensive caption was placed below the Tables with some interesting comments and clarifying the abbreviations.

C: The title of the manuscript states "What is the role of microbiome?". However, this aspect is not sufficiently disclosed by the authors, and this problem is not discussed in the Discussion section. 

A. The title was modified according to the amelioration of the Discussion section

C: These authors are not ready to significantly modify the manuscript, then from the title of the article "What is the role of microbiome?" must be deleted.

A: We tried to save the content of the Title according to the new Discussion section.

C: Comments on the Quality of English Language.The style of the English language still needs correction.

A:  WE made effort in polishing the English.

We would like to warmly thank you for these suggestions/comments that we have accepted.

Round 3

Reviewer 2 Report

Comments and Suggestions for Authors

I thank the authors for their answers.

However, the manuscript has not been substantially modified.

Unfortunately, I propose to reject the manuscript.

Comments on the Quality of English Language

Minor editing of English language required